# A large field of view two-photon mesoscope with subcellular resolution for in vivo imaging

**Nicholas James Sofroniew[1†], Daniel Flickinger[1†], Jonathan King[2], Karel Svoboda[1*]**

[1]Janelia Research Campus, Howard Hughes Medical Institute, Ashburn, United States; [2]Vidrio Technologies, Ashburn, United States

**Abstract** Imaging is used to map activity across populations of neurons. Microscopes with cellular resolution have small (<1 millimeter) fields of view and cannot simultaneously image activity distributed across multiple brain areas. Typical large field of view microscopes do not resolve single cells, especially in the axial dimension. We developed a 2-photon random access mesoscope (2p-RAM) that allows high-resolution imaging anywhere within a volume spanning multiple brain areas (∅ 5 mm x 1 mm cylinder). 2p-RAM resolution is near diffraction limited (lateral, 0.66 µm, axial 4.09 µm at the center; excitation wavelength = 970 nm; numerical aperture = 0.6) over a large range of excitation wavelengths. A fast three-dimensional scanning system allows efficient sampling of neural activity in arbitrary regions of interest across the entire imaging volume. We illustrate the use of the 2p-RAM by imaging neural activity in multiple, non-contiguous brain areas in transgenic mice expressing protein calcium sensors.

**\*For correspondence:** svobodak@ janelia.hhmi.org

[†]These authors contributed equally to this work

## Introduction

Over the last ten years cellular calcium imaging has become widely used to image activity in neuronal populations (*Grienberger and Konnerth, 2012*; *Peron et al., 2015*). Calcium imaging samples activity of all labeled neurons in an imaging volume and can readily be combined with visualization of cell type markers to analyze activity in specific cell types, the nodes of neural circuits (*Kerlin et al., 2010*; *Fu et al., 2014*; *Peron et al., 2015*).

The development of sensitive protein sensors for neuronal function has been a major driver for cellular activity imaging in vivo (*Nagai et al., 2004*; *Tian et al., 2009*; *Horikawa et al., 2010*; *Zhao et al., 2011*; *Chen et al., 2013*; *Dana et al., 2016*). More sensitive indicators allow measurement from larger numbers of neurons at fixed signal-to-noise ratios (SNR) (*Wilt et al., 2013*; *Peron et al., 2015*). Future improvements in protein sensors are expected to further increase the numbers of neurons that can be probed simultaneously.

The vast majority of cellular imaging in vivo has been done with 2-photon microscopy (*Denk and Svoboda, 1997*). 2-photon microscopy confines excitation in scattering tissue, which in turn underlies three-dimensional contrast with subcellular resolution (*Denk et al., 1994*). 2-photon microscopy demands that individual resolution elements (on the order of 1 µm$^3$) are sampled sequentially, limiting its speed. Spurred by the development of fast solid state cameras, wide-field methods such as wide-field microscopy (*Ziv et al., 2013*), light-field microscopy (*Prevedel et al., 2014*), and light-sheet microscopy (*Holekamp et al., 2008*; *Ahrens et al., 2013*) have been used to measure neural activity. Scattering and out-of-focus fluorescence rapidly degrade signal and contrast when imaging in scattering tissue with wide-field methods compared to 2-photon microscopy (*Peron et al., 2015*). As a consequence, at any point in the image, signals from multiple neurons are mixed together. Unmixing these signals to extract neural activity of individual neurons is a challenging computational

problem, which remains unsolved in general. In contrast, the high resolution and image contrast provided by 2-photon microscopy allows extraction of fluorescence signals corresponding to the activity of individual neurons and subcellular compartments (*Stosiek et al., 2003*; *Sato et al., 2007*; *Akerboom et al., 2012*; *Chen et al., 2013*; *Petreanu et al., 2012*).

Together with calcium imaging, 2-photon microscopy is now routinely used to examine behavior-related activity in populations of neurons (*Harvey et al., 2012*; *Huber et al., 2012*; *Peron et al., 2015*). The majority of studies image only dozens to hundreds of neurons in one brain region at a time. The advent of sensitive protein indicators for calcium (*Nagai et al., 2004*; *Tian et al., 2009*; *Horikawa et al., 2010*; *Zhao et al., 2011*; *Chen et al., 2013*) and transgenic animals expressing these indicators (*Dana et al., 2014*; *Madisen, 2015*) promises much higher throughput. 2-photon microscopy has already been used together with GCaMP6 (*Chen et al., 2013*) to measure activity in approximately 1800 pyramidal neurons at 7 Hz using 50 mW of power at the sample (*Peron et al., 2015*). Optimization of the delivered laser power and 2-photon excitation efficiency will allow tracking of more than 10,000 pyramidal neurons at similar signal-to-noise ratios and sampling frequencies. This opens up the possibility of tracking representative neuronal populations across multiple brain regions.

Even relatively simple animal behaviors involve multiple brain regions, which are often non-contiguous (*Hernández et al., 2010*; *Guo et al., 2014*). Simultaneous tracking of activity in these brain areas is not possible using standard microscopes, because mechanical movement of microscopes or specimens is slow. High-resolution microscopes have fields of view (FOV) that are smaller than most brain areas, whereas large FOV microscopes have small numerical apertures and cannot resolve individual neurons. To bridge the gap between single cells and brain regions we developed a mesoscale 2-photon microscope capable of imaging a FOV of 5 mm, with subcellular resolution. We use the term 'mesoscale microscopy' to refer to imaging with subcellular resolution and fields of view spanning multiple brain areas (several millimeters).

We implemented fast lateral and axial scanning to both maximize the number of neurons that can be sampled nearly simultaneously and the flexibility in choosing the imaged neurons. The fluorescence signal collected per unit time is ultimately limited by the average power that can be delivered to the specimen, which in turn is limited by heating. In the mouse brain the maximum allowed power is approximately $P_{max}$ = 200 mW (*Podgorski and Ranganathan, 2016*). The signal rate for one scanning beam is $S_1 = a\,P_{max}^2$, where $a$ is a parameter that depends on the illumination, the experimental preparation (e.g. the type and concentration of the fluorescent probe), and fluorescence detection. In another popular scheme the excitation beam is partitioned into $n$ beams for multiplexed excitation (*Lecoq et al., 2014*; *Okun et al., 2015*). Dividing the power into $n$ laser foci for multiplexed imaging, the signal is $S_n = a\,n\,(P_{max}/n)^2$. This division corresponds to reduced overall signal levels (by $1/n$) and a corresponding reduction in the number of neurons sampled, at fixed signal-to-noise ratio. Increasing the power of each individual beam beyond a total (summed) power of $P_{max}$ = 200 mW would cause heating and associated damage. Given the relatively slow dynamics of calcium-related fluorescence (*Peron et al., 2015*), serial scanning with a single laser focus maximizes the number of neurons that can be probed.

Several 2-photon microscopes have been designed for large field of view in vivo imaging. Tsai et al designed an elegant scanning system based on off-the-shelf components, producing a scan field of 10 mm (*Tsai et al., 2015*). Stirman et al used a custom objective (*Stirman et al., 2014*) with a field of view of 3.5 mm. Neither system provides subcellular resolution in the axial dimension (14 µm and 12 µm, respectively), implying that signals from multiple neurons and neuropil may be conflated. In current implementations these instruments use standard galvo mirrors, which are too slow for some types of functional imaging. These microscopes also do not implement rapid axial scanning, which is critical for targeting specific neurons in different parts of the FOV. In addition, the microscopes are corrected for aberrations at only one excitation wavelength (800 nm and 910 nm respectively), limiting their use to a subset of fluorescent probes. A large FOV (5 mm) mesolens for confocal microscopy, with numerical aperture 0.47, has also been reported (*McConnell et al., 2016*).

We have developed a 2-photon random access mesoscope (2p-RAM), optimized for imaging populations of neurons in multiple brain regions during behavior. 2p-RAM provides diffraction-limited resolution in a cylindrical volume (∅ 5 mm x 1 mm). It is fully corrected for excitation light in the 900–1070 nm range, allowing 2-photon imaging of most widely used fluorescent proteins. 2p-RAM

is based on a fast three-dimensional scanning system that allows random access to hundreds of thousands of neurons for targeted interrogation.

## Results

We developed a 2-photon random access mesoscope (2p-RAM) with subcellular resolution. We first outline the key specifications, which correspond to specific engineering challenges. We then explain the optical path of the mesoscope, followed by a brief description of the design process. Finally we illustrate the mesoscope performance in calibration experiments and imaging neural activity in vivo.

### Key specifications

First, the diameter of the FOV should be five millimeters. This size allows sampling of most cortical regions in the mouse brain that are nearly coplanar. For example, it would be possible to image the primary somatosensory cortex, the primary motor cortex and parts of secondary motor cortex nearly simultaneously (*Figure 1A*). Second, the mesoscope should have better than cellular resolution, implying axial resolutions substantially smaller than the typical size of a soma (diameter, 10 μm) (*Peters and Kaiserman-Abramof, 1970*). We specified a numerical aperture (NA) of 0.6, which translates to approximately 0.61 μm lateral resolution and 4.25 μm axial resolution (for $\lambda$ = 970 nm; *Figure 1B*) (*Zipfel et al., 2003*). Third, the mesoscope should produce diffraction-limited performance and high two-photon excitation efficiency over the range of $\lambda$ = 900–1070 nm. This spectral range corresponds to fluorescent protein sensors based on GFP and various red fluorescence proteins (*Akerboom et al., 2013*). Fourth, the mesoscope should maximize the collection of fluorescence signal, ideally with a large collection NA. We specified a collection NA of 1.0 (specifications list in *Figure 1C*).

Fifth, a key challenge for mesoscale 2-photon microscopy is efficient laser scanning. Faster scanning provides better time resolution for measurements of neural activity, reduced photodamage, and tracking larger numbers of neurons. Rapid scanning methods rely on resonant scanning mirrors (*Fan et al., 1999*) or acousto optic deflectors (AODs) (*Reddy and Saggau, 2005*), both of which have scan angles on the order of a few degrees, corresponding to relatively small (several hundred μm) scans in the specimen plane. To access the entire scan field rapidly we designed a lateral scan unit with multiple scanners in series. A fast resonant scan is moved over the specimen in a flexible manner using a galvanometer scanner, allowing rapid sampling of activity in widely dispersed brain regions (*Figure 1D*).

The sixth and final challenge is scanning along the optical axis of the microscope. In most scanning microscopes, focusing is achieved by moving the objective. Because of the mechanical inertia of the bulky objective, focusing is more than two orders of magnitude slower than lateral scanning (*Göbel et al., 2007*). This limit is unacceptable for mesoscale imaging because the structures of interest to be sampled in different brain regions are typically in focal planes that differ by up to several hundred micrometers (*Figure 1E*). We addressed this problem by implementing a rapid remote focusing unit in which a light-weight mirror is moved instead of the objective (*Botcherby et al., 2012*). Rapid remote focusing also allowed us to relax flatness of field specifications, because rapid focusing can partially correct for curved scan fields online.

### Implementation

#### Overview

The 2p-RAM was assembled on a vertically mounted breadboard (*Figure 2A,C,E*). The refractive optics of the microscope are predicted to introduce 25,000 fs$^2$ group-delay dispersion (GDD) (at $\lambda$ = 1000 nm). GDD will cause the light arriving at the focus to be spread out in time, resulting in reduced efficiency for two-photon excitation (*Denk et al., 1995*). To reduce GDD at the sample, the Ti:Sapphire laser beam first passed through a prism-based GDD compensation unit (*Akturk et al., 2006*) (*Figure 2B*). With GDD compensation enabled, we measured a pulse width of 106 fs at the sample using an autocorrelator (Carpe; APE) (see *Figure 2—figure supplement 1*), comparable to the pulse width emitted by our laser (Mai Tai HP; Spectra Physics).

The microscope is motorized to allow flexible access to the specimen: it moves in x, y (>50 mm travel); it rotates around an axis that passes through the specimen plane and is parallel to the long

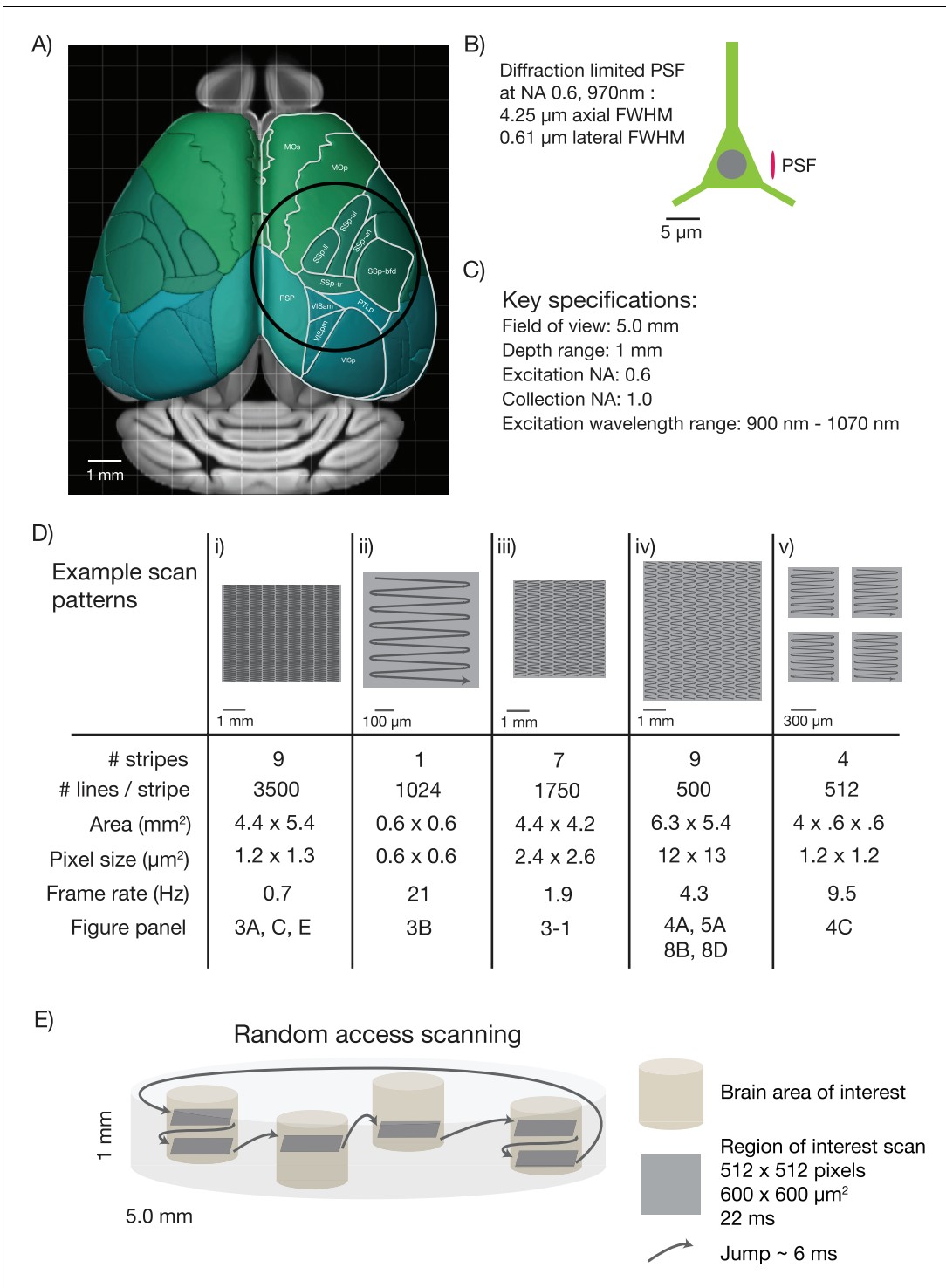

**Figure 1.** Specifications for the 2p-RAM. (**A**) Schematic of the dorsal surface of the mouse cortex (from Allen Brain Atlas, Brain Explorer 2). The overlaid circle corresponds to the specified FOV (∅ 5 mm). MOp: primary somatomotor, MOs: secondary somatomotor, SSp-ul: primary somatosensory upper limb, SSp-ll: primary somatosensory lower limb, SSp-un: primary somatosensory unassigned, SSp-tr: primary somatosensory trunk, SSp-bfd: primary somatosensory barrel field, PTLp: posterior parietal association, RSP: retrosplenial, VISam: anteromedial visual, VISpm: posteromedial visual, VISp: primary visual. (**B**) Schematic of a neuron (approximate soma diameter, 10 μm), and a diffraction limited point spread function (PSF) corresponding to NA 0.6 and 970 nm. Subcellular resolution implies that the PSF must be contained within a single neuron. (**C**) Key specifications for the
*Figure 1 continued on next page*

*Figure 1 continued*

mesoscope. (**D**) Imaging parameters used in the experiments. In all cases a fast resonant scan (time per scan line, 42 μs; 24 kHz line rate) is made over patches 608 μm wide. The time per patch is 42 μs x number of lines. The entire FOV can be covered in nine stripes. A slow (0.7 Hz), high-resolution scan with 3500 lines per stripe was used to cover a large portion the FOV (i). A fast (4.3 Hz), low-resolution scan across the entire scan range has up to 500 lines per stripe (iv). More typical scans will involve sampling multiple smaller patches at high resolution and frame rates (v). See implementation for discussion of the scanning patterns. (**E**) Schematic showing a typical use-case. The entire imaging volume is a cylinder with a 5 mm diameter and 1 mm depth. The fast scan is made over patches in different parts of the volume.

axis of the microscope ( ± 20 degrees travel); it moves up and down along the beam axis for coarse focusing (travel +38 mm−13 mm). Since the microscope moves with respect to the laser, the beam was threaded into the microscope through a multi-stage periscope.

## Remote focus unit

Within the core of the microscope, the beam first enters the remote focus (RF) unit through a polarizing beam splitting cube and a quarter wave plate. The RF unit consists of a custom RF objective, and a small mirror (mass, 170 mg; PF03-03-P01, Thorlabs) mounted on a voice coil (LFA 2010 with a sca814 controller; Equipment Solutions). The beam passes through the RF objective and is reflected by the mirror back through the RF objective. The quarter wave plate and polarizing beam splitting cube together then direct the beam into the lateral scan unit. The coatings of the RF objective were optimized for high transmission in the IR range. These coatings and the small number of elements (5) combined to give high transmission through the RF objective (97%).

The entrance aperture of the RF objective is conjugated to the entrance aperture of the imaging objective. The RF mirror is approximately conjugate to the focus in the specimen plane. Axial movement of the RF mirror changes the axial location of the focus in the specimen (scale factor, 1.23; i.e. a 100 μm move of the RF mirror causes a 123 μm move of the focus). Depending on the axial position of the RF mirror, the wavefront entering the imaging objective will be converging, parallel, or diverging. For standard microscope objectives, which obey the sine condition, only parallel rays will converge to a diffraction-limited point in the specimen (*Born and Wolf, 1980*). Spherical wavefronts in the objective pupil will produce spherical aberration at the focus. Rays entering the objective back aperture at its periphery will focus to a different point along the axis, thereby producing a larger focal volume with lower peak intensity. These spherical aberrations degrade resolution and contrast. The RF objective was designed to counteract the spherical aberrations produced by the imaging objective, so that diffraction-limited imaging can be achieved throughout the microscope's imaging volume (*Botcherby et al., 2012*). The imaging objective also has axial chromatic aberration that varies with RF depth, which is compensated by the RF objective. The RF unit allows rapid axial movement of the focal plane, since the RF mirror can be moved much faster than the objective or the specimen.

## Scanning system

We designed a lateral scanning unit to efficiently scan arbitrary regions of interest in the FOV of the microscope. A resonant scanner produces a fast line scan (24 kHz line rate) (CRS 12 K, Cambridge Technology). Because of the limited angular range of the scanner (10 degrees, optical peak-to-peak) this results in a scan range of 0.6 mm or less in the specimen plane. This scan line is moved over the specimen using a pair of galvo scanners.

Microscopes using remote focusing have to deal with non-planar wavefronts in their pupil locations. Maintaining conjugation between all scan mirrors and optic pupils is a critical issue for these microscopes. Otherwise, the movement of any scan mirror that is not conjugated to a subsequent optic pupil, such as the entrance pupil of the objective, will cause a lateral shift of the wavefront at that pupil. Wavefront non-planarity and lateral shift can cause significant optical aberrations.

The beam is directed through a pupil relay (PR1; magnification, 0.44) into the resonant scanner (open aperture, 4.8 mm), and then through a second identical pupil relay in reverse (PR2; magnification, 2.3) into a virtually conjugated galvo pair (VCGP). The VCGP unit conjugates the x and y scan

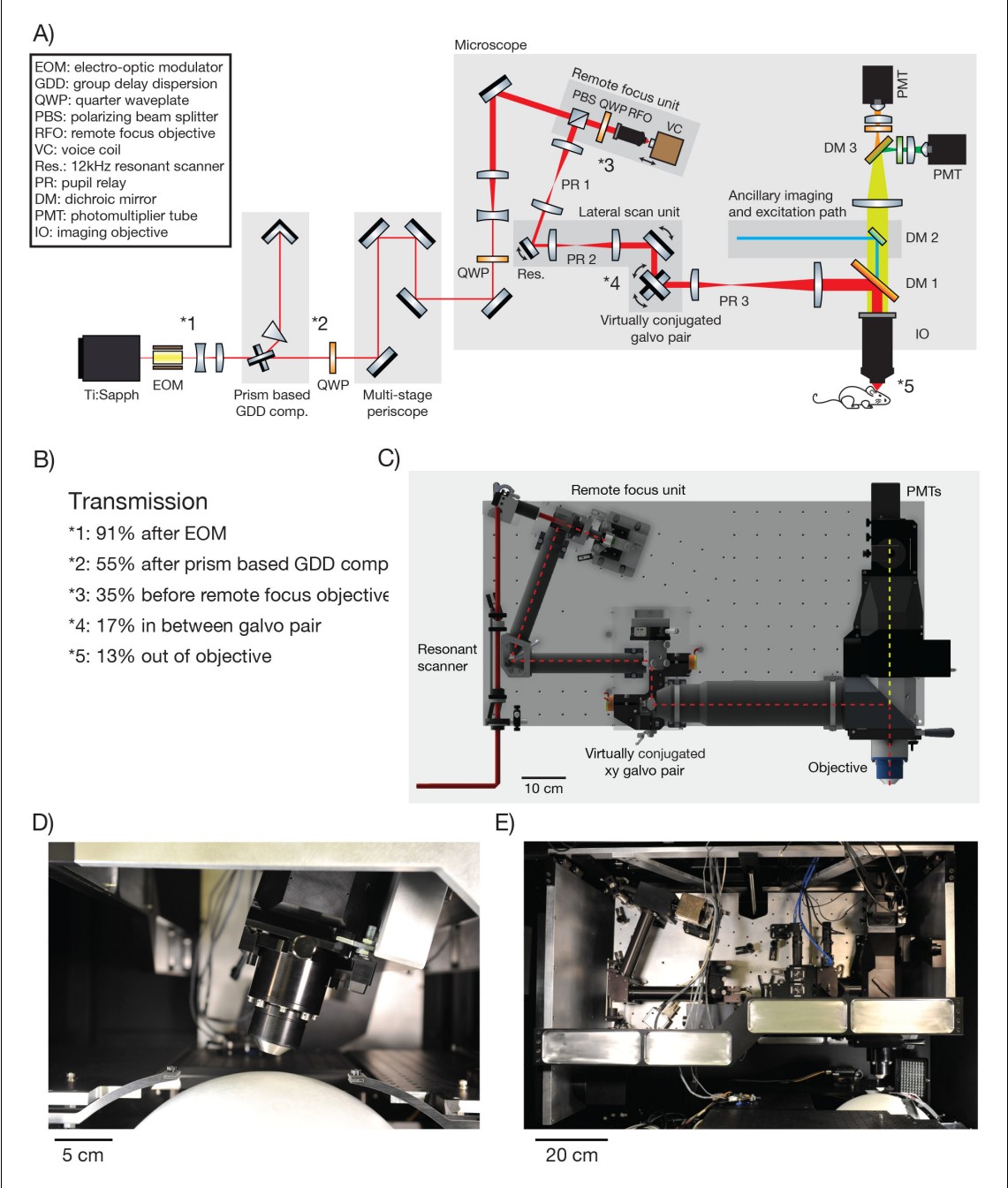

**Figure 2.** 2p-RAM optics and optomechanics. (**A**) Optical layout. Several routing mirrors were omitted for clarity. See text for detailed description. (**B**) Power transmission measurements at 970 nm. (**C**) CAD model of the microscope. The microscope was assembled on a vertically mounted breadboard measuring 1.0 m x 0.5 m. The objective is 63 mm wide, 120 mm long, and weighs 1.43 kg. (**D**) Photograph of microscope objective. (**D**, **E**, photocredit; Matt Staley). (**E**) Photograph of vertically mounted breadboard.

The following figure supplements are available for figure 2:

**Figure supplement 1.** Pulse width measurements.
**Figure supplement 2.** Distances between the critical 2p-RAM components.

galvo pair without a large-aperture relay made from refractive optics. Alternatively conjugation could have been accomplished using a custom pupil relay using refractive optics. However, due to its large size, this relay would be the most complex and expensive part of the microscope. The VCGP provides a compact and cost-effective solution.

The VCGP uses three galvo mirrors (rather than two) to ensure that the pupils of the x and y scanning galvos are coincident. The first two mirrors (open aperture, 20 mm; Cambridge Technology 6M2420X40B025S4 mirror on 6240HM40B galvo) are moved in the same plane so that the beam rotates about a pupil centered on the third mirror (open aperture, 14 mm, Cambridge 6M2314X44B025S4 mirror on 6231HM44C galvo), which rotates in the orthogonal direction. In this manner the pupil remains substantially stationary over a large scan range. The 20 mm, virtually conjugated galvo mirrors each have a scan range of 40 deg (optical peak-to-peak, OPTP). However, the virtual conjugate arrangement of these mirrors results in a total scan range of 32.2 deg OPTP at the following pupil. The 14 mm galvo mirror has a scan range of 44 deg OPTP. The 20 mm mirrors can perform a step across their entire range with step-and-settle time of 3.4 ms. Smaller steps with these mirrors, and all steps with the 14 mm mirror, are faster.

## Objective

The exit pupil of the VCGP is imaged by another pupil relay (PR3; magnification, 2.3) onto the entrance pupil (diameter, 25.6 mm) of the imaging objective (NA, 0.6, focal length 21 mm) (*Figure 2D*). The excitation beam is routed into the imaging objective using a shortpass dichroic (DM1; size 70 x 99 mm; 750 nm shortpass custom coating design from Alluxa). The front lens has a tip diameter of 15 mm. The distance between the edge of the front lens and the top of the specimen is 2.7 mm. Because remote focusing moves the focal plane into the specimen without movement of the objective or specimen, the effective working distance of the objective is larger than 3 mm. In vivo imaging is usually performed through a microscope coverglass (*Trachtenberg et al., 2002*; *Holtmaat et al., 2009*). The optical system is corrected to operate in a diffraction-limited manner through a # 4 coverglass (450 µm of BK7 glass; equivalent to a stack of three # 1 coverglasses). Power transmission through the entire system is 13% (*Figure 2B*).

## Fluorescence collection path

Fluorescence light passes through the primary dichroic (DM1) into the detector arm of the microscope. An optional second dichroic mirror (DM2) allows for an ancillary optical path for one-photon imaging and photostimulation. A third dichroic mirror (DM3) divides the fluorescence path into two signals, each directed into one of two GaAsP photomultiplier tubes (PMTs) (Hamamatsu H11706-40).

In a 2-photon microscope, the etendue of fluorescence light collection by the objective lens must be matched or exceeded by the etendue of the light detector. This condition is critical for optimal detection of precious signal photons. For most microscopes this condition is not an issue, as PMTs have photocathodes that are much larger than the objective FOV, as well as sizable collection angles (*Tsai et al., 2002*). For the 2p-RAM the imaging objective collects emitted light with an NA of 1.0, and a collection FOV of 6 mm (*Oheim et al., 2001*). The diameter of the imaging objective pupil relevant for fluorescence detection is 42 mm. The etendue of this collected emission light exceeds the etendue of the current state-of-the-art PMTs which have a photocathode diameter of 5 mm and a usable NA of approximately 0.9 (unpublished measurements). This makes it impossible for all of the fluorescence light collected by the objective to be directed onto the PMT using conventional imaging schemes (*Mainen et al., 1999*; *Tsai et al., 2002*). To overcome this problem we used oil to directly couple a custom-made condenser lens to the glass enveloping the PMT. This allowed the collection NA of the PMT to reach 1.4. In this manner all of the fluorescence that is collected by the imaging objective is projected onto the PMT for detection (not counting reflection losses).

Our collection path does not image the back pupil of the objective onto the PMT (*Denk et al., 1995*; *Mainen et al., 1999*), as that design would require significantly more complex collection optics. As a result, the spatial distribution of the signal on the PMT photocathode varies with location of the signal source within the sample. The spot size of the projected light from a point source in the sample on the photocathode varies from 1.5 mm to 2.7 mm, depending on the location of the source point. The microscope image is therefore sensitive to inhomogeneities in the photosensitivity

of the PMT photocathode. For the most critical applications it may be necessary to select PMTs with homogeneous photocathodes or employ computational methods to correct for the inhomogeneity.

## Optical design

The excitation path, including the RF objective, pupil relays 1–3, and imaging objective, was modeled in silico (Zemax). We searched for configurations of optical elements with diffraction-limited performance and high two-photon excitation efficiency across the ∅ 5x1 mm imaging volume, for excitation wavelengths over the range 900–1070 nm. Performance was assessed using the Strehl ratio (SR), which is a measure of the quality of optical image formation. Without aberrations an optical system has SR = 1; aberrations drive SR toward zero. We optimized for SR > 0.8 across the imaging volume, a commonly used criterion for diffraction-limited performance.

Two-photon excitation efficiency depends on the propagation time delay difference (PTD) for different rays across the pupil of the system (*Estrada-Silva et al., 2009*). For PTD not much less than the pulse duration, the efficiency of two-photon excitation would be severely reduced. PTD was kept to less than 50 fs (rms). As far as we know no other microscopy system has been designed with optimization of the PTD as a design criterion.

Other design parameters included: The number, size and complexity of the optical elements; focusing range of the RF unit; nature and cost of the optical glasses. Field curvature was limited to less than 160 μm over the 5 mm FOV. Finally, tolerancing was performed to minimize sensitivities and guide manufacturing (Code V). No stock lenses were used for these modules, as doing so in most cases would not allow sufficient control over the tolerances to meet performance specifications. The element count for each module was: RF objective, 5; pupil relay 1/2 (identical), 4; pupil relay 3, 6; imaging objective, 6. (The modules were manufactured by Jenoptik. Jupiter, Fl; part numbers RF objective: 14163200, pupil relay 1/2: 14506000, pupil relay 3: 14163100, imaging objective: 14163000; for a table of critical distances between these elements necessary for assembly see *Figure 2—figure supplement 2*). A detailed description of the custom optics and the design process will be published elsewhere.

## Calibration experiments

### Brightness across the field of view

We measured the signal brightness across the FOV. In a homogeneous sample, such as a solution of fluorophore, image brightness is determined by the two-photon excitation efficiency and the collection efficiency for fluorescence, both of which depend on position in the FOV. We acquired an image of a 1 mm thick bath of fluorescein (λ = 970 nm). At an imaging depth of 500 um, brightness was constant up to approximately 2.5 millimeters from the center, beyond which image brightness collapsed (*Figure 3*). The image showed inhomogeneities across the resonant scan (608 μm), apparent as shading in the vertical stripes in *Figure 3A*. The shading is produced by wave front error introduced by pupil relay 2 during rapid resonant scanning. Smaller resonant scans would reduce this shading. Inhomogeneities in brightness within the specified FOV were on the order of 30% (*Figure 3C*), including a small dip in brightness in the center of the field (this dip is caused by spatial inhomogeneity in the PMT sensitivity across its photocathode; see below).

### Resolution

We next measured the resolution of the microscope. Fluorescent microbeads (0.5 μm; Fluoresbrite Calibration Grade YG microspheres CAT# 18859, Polysciences Inc.) were embedded in agar (1.5%; Agarose, Type III-A Sigma A9793) at a dilution of 1:10,000. Three-dimensional images stacks containing images of beads were collected at various positions in the imaging volume (λ = 970 nm). Resolution was estimated based on the lateral and axial full-width-at-half-max (FWHM) of the bead images (*Figure 4*). The lateral FWHM of the bead image was 0.66 ± 0.003 μm (SEM) and the axial FWHM was 4.09 ± 0.06 μm (SEM). Approximately 2 millimeters from the center of the FOV, the resolution started to degrade gradually (*Figure 4I,J,K*). At the edge of the FOV (2.5 mm from the center) the lateral FWHM was 0.89 ± 0.006 μm (SEM), and the axial FWHM 6.88 ± 0.07 μm (SEM). At the center of the FOV, performance was diffraction-limited up to 1 mm deep into the sample (*Figure 4—figure supplement 1*). This performance was similar at different positions in the field and different imaging depths up to 1 mm into the sample (*Figure 4—figure supplement 2*). Resolution was

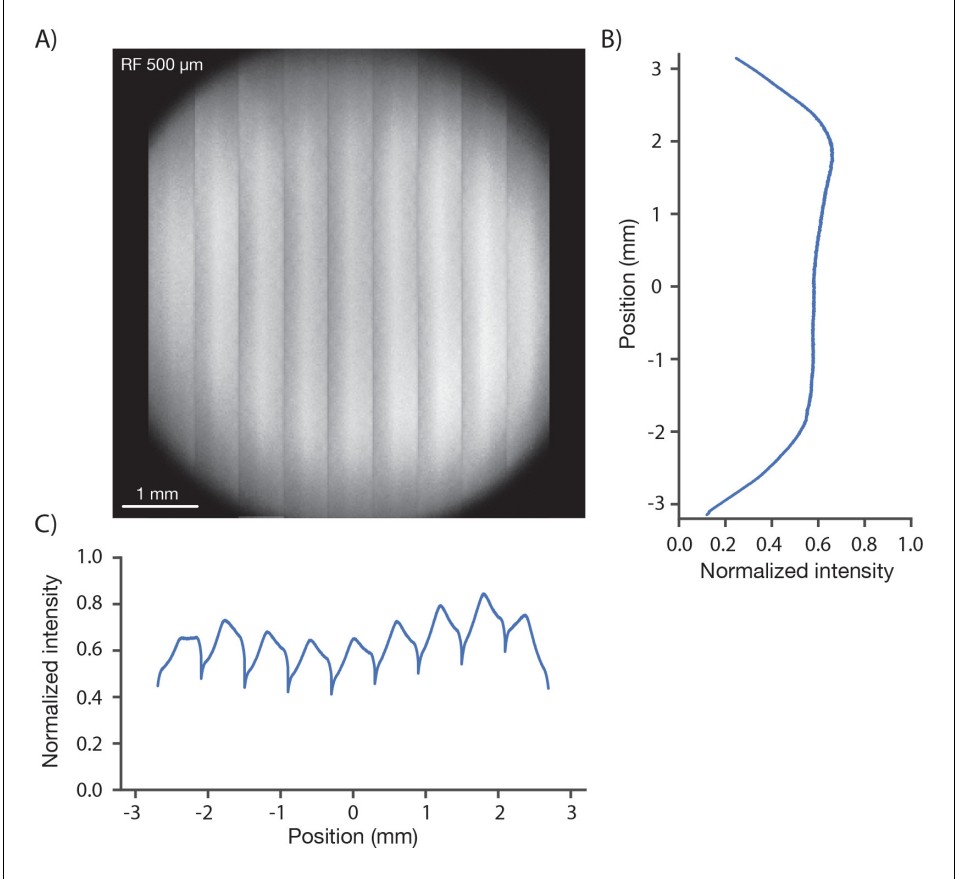

**Figure 3.** Image brightness. (**A**) Image of a uniform fluorescent sample (fluorescein solution, Sigma F6377, 100 µM, 1 mm thick under 450 µm of coverglass, average of 30 frames). The image was acquired in 9 stripes, each approximately 600 µm wide (see **Figure 1D, iv** for details). The intensity variations in the image are caused by changes both in excitation and collection efficiency across the field. (**B**) Y profile (along slow scan axis) through slice at the central RF position normalized to the peak signal, which was located off the main two axes. (**C**) X profile (along fast scan axis) profile through central slice. The dips in signal at the edge of the stripes are caused by aberrations at large resonant scan angles.

consistent across wavelengths from 900 nm to 1040 nm (the limit of the tuning range of our Ti:Sapphire laser) (**Figure 4—figure supplement 3**). Resolution also remained essentially unchanged for small (up to ± 2.5°) tilts of the coverglass with respect to the beam axis (**Figure 4—figure supplement 4**). We conclude that the 2p-RAM is capable of subcellular resolution over the entire imaging volume (∅ 5x1 mm cylinder) and is robust to moderate misalignments.

The brightness of the imaged beads dropped off towards the edge of the FOV (up to a factor of four 2.5 mm from the center of the FOV; **Figure 4K**). This is consistent with the observed increase in the size of the PSF (**Figure 4J**). In 2-photon microscopy, the brightness of objects that are on the order of, or smaller than the diffraction limit are highly sensitive to degradation in the PSF. Decreased fluorescence collection efficiency at the edges of the FOV could also contribute to dimmer images at the edge of the FOV. Conversely, resolution was consistent across the center of the field, (**Figure 4I,J**) suggesting that the dip in brightness (**Figure 3**) at the center of the field is likely due to a dip in PMT sensitivity in the center of the FOV (see below).

## Collection efficiency

Biological 2-photon microscopy is typically limited by photodamage produced by excitation light. Better detection of fluorescence signal allows reduced excitation for fixed SNR. Optimizing fluorescence detection is therefore a critical aspect of designing a 2-photon microscope for in

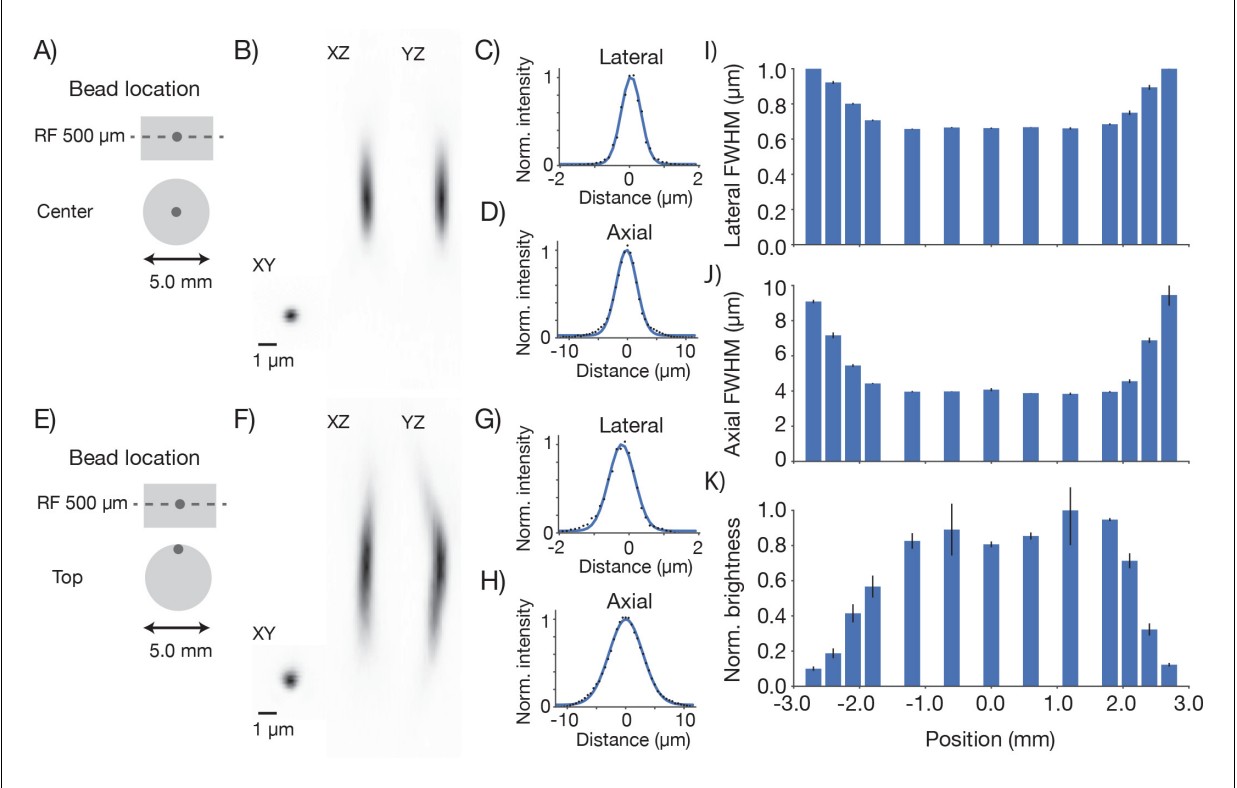

**Figure 4.** Point spread functions. Point spread functions were measured at 13 different positions in the field at the middle depth of the imaging volume. The sample consisted of fluorescent microbeads (diameter, 0.5 μm, embedded in 1 mm thick 1.5% agar, under 450 μm of coverglass) imaged at 970 nm. The excitation intensity was 8.75 mW, below excitation saturation of the beads. At each position, a 61 μm x 61 μm x 35 μm volume (average of 30 acquisitions) was acquired with 8.4 x 8.4 x 2 pixels per μm. Bead that were separated by at least 2 μm x 2 μm x 12 μm from the edges of the volume and other beads were analyzed. 2–13 PSFs per volume were measured. (**A**) Example measurement at the center of the FOV. (**B**) Mean intensity projections through the XY, XZ, and YZ slices of the bead. (**C**) Gaussian fit through the lateral slice through the bead image. (**D**) Same as (**C**), axial slice. (**E–H**) Same as (**A–D**) for a bead at the edge of the FOV (2.4 mm from the center on the top). (**I**) Average lateral full-width-at-half-max (FWHM) of the central XY slice of a bead, averaged across the X and Y directions. (**J**) Average axial FWHM. (**K**) Average normalized brightness of the beads. Error bars, standard deviation.

The following figure supplements are available for figure 4:

**Figure supplement 1.** Point spread functions across imaging depth Point spread functions were measured at five different depths at the center of the FOV.

**Figure supplement 2.** Point spread functions across the imaging volume Point spread functions were measured at 15 different locations across the imaging volume.

**Figure supplement 3.** Point spread functions at different excitation wavelengths.

**Figure supplement 4.** Point spread functions as a function of coverglass angle.

vivo imaging. In scattering samples the objective numerical aperture and its FOV contribute to fluorescence detection (*Oheim et al., 2001*). We therefore characterized the signal collection efficiency of our system as a function of position of the signal source and numerical aperture (*Figure 5*).

We separately varied the position of a light source and its NA while measuring the signal detected by the PMT (*Figure 5A*). A pinhole was mounted in front of the light source (LED) and imaged using an imaging system into the specimen plane. The NA of the light cone was controlled using an aperture in an intervening aperture plane. The microscope was moved on a 0.5 millimeter grid around this assembly.

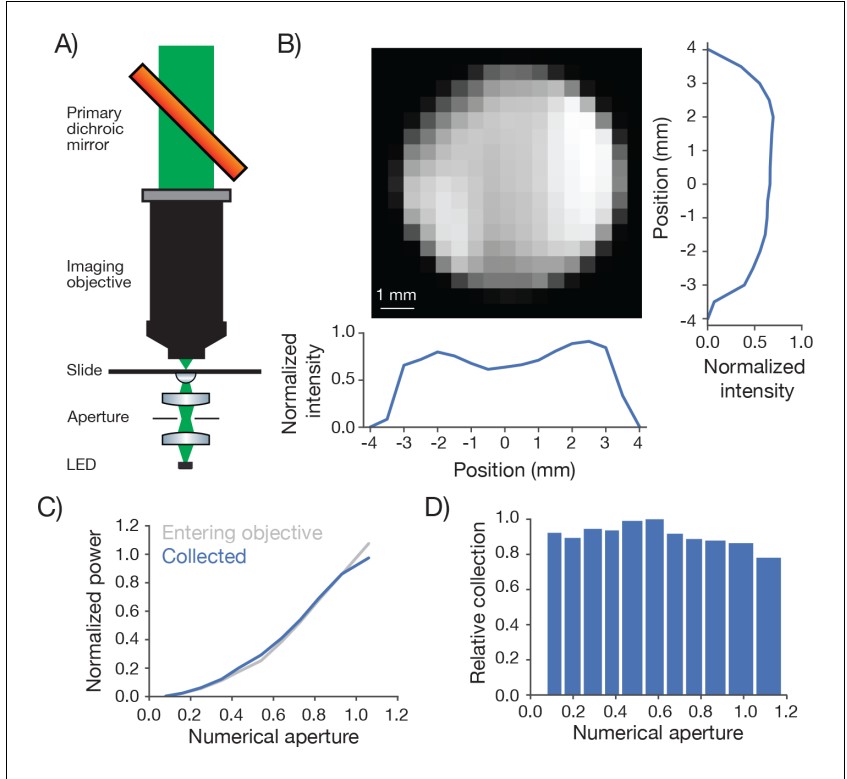

**Figure 5.** Measurement of collection efficiency. (**A**) Apparatus for measuring signal collection across the FOV and determining the NA of the collection optics. The light source consisted of a LED (530 nm) and adjacent pinhole, with a miniature optical relay that projected the pinhole onto the top of a microscope slide. The projected spot had a 460 μm diameter. An aperture in the relay could be adjusted to change the NA of the light source from 0 to 1. (**B**) Normalized signal intensity measured with the PMT (not shown in **A**) as a function of position of the light source. The objective was moved in a 0.5 mm grid relative to the light source. Right, Y slice through the center of the image. Left, X slice through the center of the image. The peak signal was located off the main two axes. (**C**) Signal intensity measured by the PMT as a function of the numerical aperture of the light source (blue) overlaid on the power entering the objective (gray). As the NA of the light source increases the power entering the objective scales approximately as the square of the NA, as expected. The power entering the objective was measured with a high NA accepting power meter (Thor; S170C). Measurement was taken with the 460 μm spot in the center of the field. (**D**) Peak normalized ratio of the intensity measured by the PMT to the power entering the objective as a function of the NA of the light source (axis sampled non-uniformly). The signal intensity measured through the collection optics scales with the power entering the objective until it drops off around NA 1.0, significantly larger than the 0.6 excitation NA of the objective.

The PMT signal was remarkably constant across the FOV and beyond (*Figure 5B*). The signal showed a modest dip in the center of the FOV (approximately 20% reduced sensitivity). This dip was tracked down to inhomogeneous sensitivity on the PMT photocathode (data not shown). This dip also likely accounts for the lower brightness at the center of the field (*Figure 3*).

We next changed the NA of the light source. The total light entering the objective was roughly proportional to the second power of the NA (*Figure 5C*). The signal detected by the PMT was proportional to the signal entering the objective up to NA 1.0, the specified NA for fluorescence detection of the imaging objective (*Figure 5C,D*). We conclude that our fluorescence collection system performs nearly optimally across the specified FOV (∅ 5 mm). The overall transmission of the collection path, including reflection and transmission losses, was 71%.

## Field curvature and its correction using remote focusing

To characterize the field curvature of the microscope we imaged a thin and flat fluorescent sample (*Figure 6A*). If the scan transects the sample the fluorescent image is a circle (*Figure 6B*). The radius

of the circle increases monotonically as the RF is adjusted downward. In this way we mapped the field curvature of the microscope. The focal position changed by up to 158 μm at 2.5 mm from the center of the FOV.

We implemented a field curvature correction by the RF unit in our scan control software. Following a 300 μm step, the RF mirror settles to better than 2.5 μm in 6.77 ± 0.02 ms (SEM) (*Figure 6— figure supplement 1*). The correction was done on a line-by-line basis, using the average position of the resonant mirror during the line as the position to correct. This scheme corrects the field curvature associated with movements of the slower galvo mirrors (i.e. along the vertical image stripes). Field curvature along the resonant scan could not be corrected in this manner because the RF unit does not have the necessary bandwidth. Not correcting the field curvature along the resonant axis (*Figure 6C*, bottom) causes striping artifacts present in the images (*Figure 7*).

## In vivo imaging

We next imaged neurons in transgenic mice expressing fluorescent proteins. Mice were prepared with a stainless-steel head plate and a D-shaped cranial window using standard surgical procedures (*Peron et al., 2015*). The D-shaped window was 450 μm thick and made from three layers of custom cut coverglass (Potomac Photonics). The window spanned from the lambda suture to 3 mm anterior of Bregma and from the midline 5 mm laterally, allowing optical access to a large fraction of the cortical surface on the left hemisphere.

Low magnification, high-resolution images (λ = 970 nm) were acquired in anesthetized mice expressing GCaMP6f in a subset of pyramidal neurons (GP5.17 mice) or GCaMP6s in the vast majority of pyramidal neurons (GP4.3) (*Dana et al., 2014*). Superficial images showed the vascular pattern on the surface of the brain (GP5.17, *Figure 7A,B*; GP4.3, *Figure 7C*, *Video 1*). Higher magnification views revealed single neurons and their dendrites (*Figure 7B*). Resolution and contrast degrade with imaging depth in vivo, caused by aberration and scattering (*Ji et al., 2012*). However, images of neurons appeared as ring-like structures up to 600 μm deep in the tissue (*Video 1*), reflecting GCaMP exclusion from the nucleus (*Tian et al., 2009*) and demonstrating subcellular resolution of the 2p-RAM.

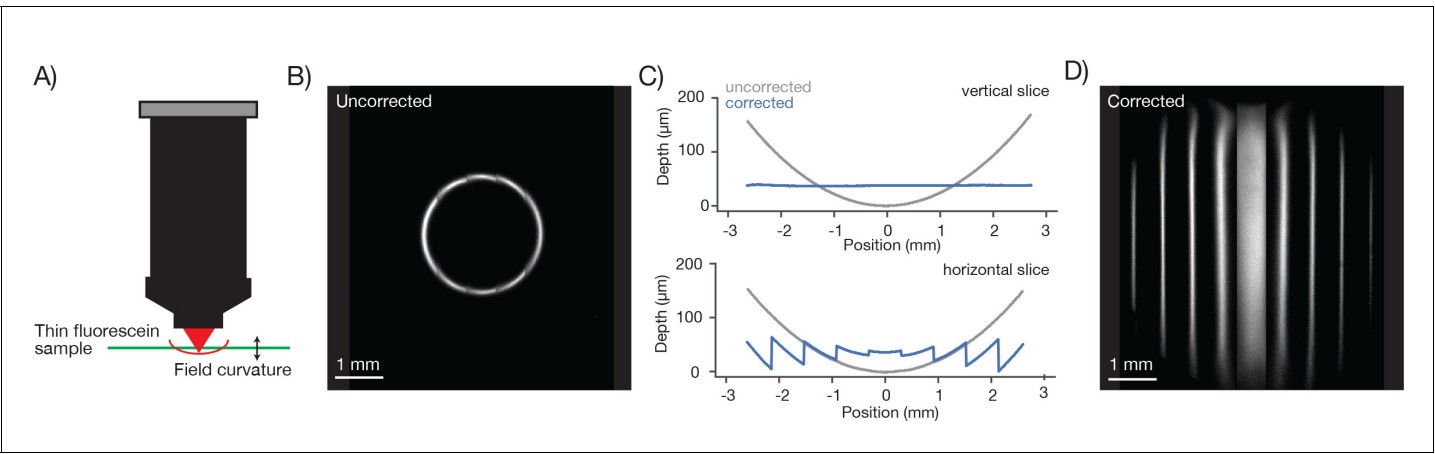

**Figure 6.** Characterization and correction of field curvature. (A) Field curvature was measured using a thin fluorescein sample (11.2 μm, sandwiched between a microscope slide and 450 μm of coverglass, imaging conditions as in *Figure 3A*). At a particular RF position the image of the sample was a fluorescent ring. We measured the diameter of the ring as a function of RF position. (B) Image of the sample without any correction for field curvature. The sample appears as a thin ring. (C) Measured field curvature (gray: before field curvature correction, blue: after field curvature correction). (D) The RF mirror was programmed to compensate for the field curvature. The compensation is done on a line-by-line basis using the average position of the resonant mirror during the line as the point to correct. This compensation is able to correct the field curvature along each stripe well (top), but is unable to correct the field curvature within a resonant scan as the resonant mirror moves too fast (bottom).

The following figure supplement is available for figure 6:

**Figure supplement 1.** Characterization of remote focus mirror positioner (voice coil).

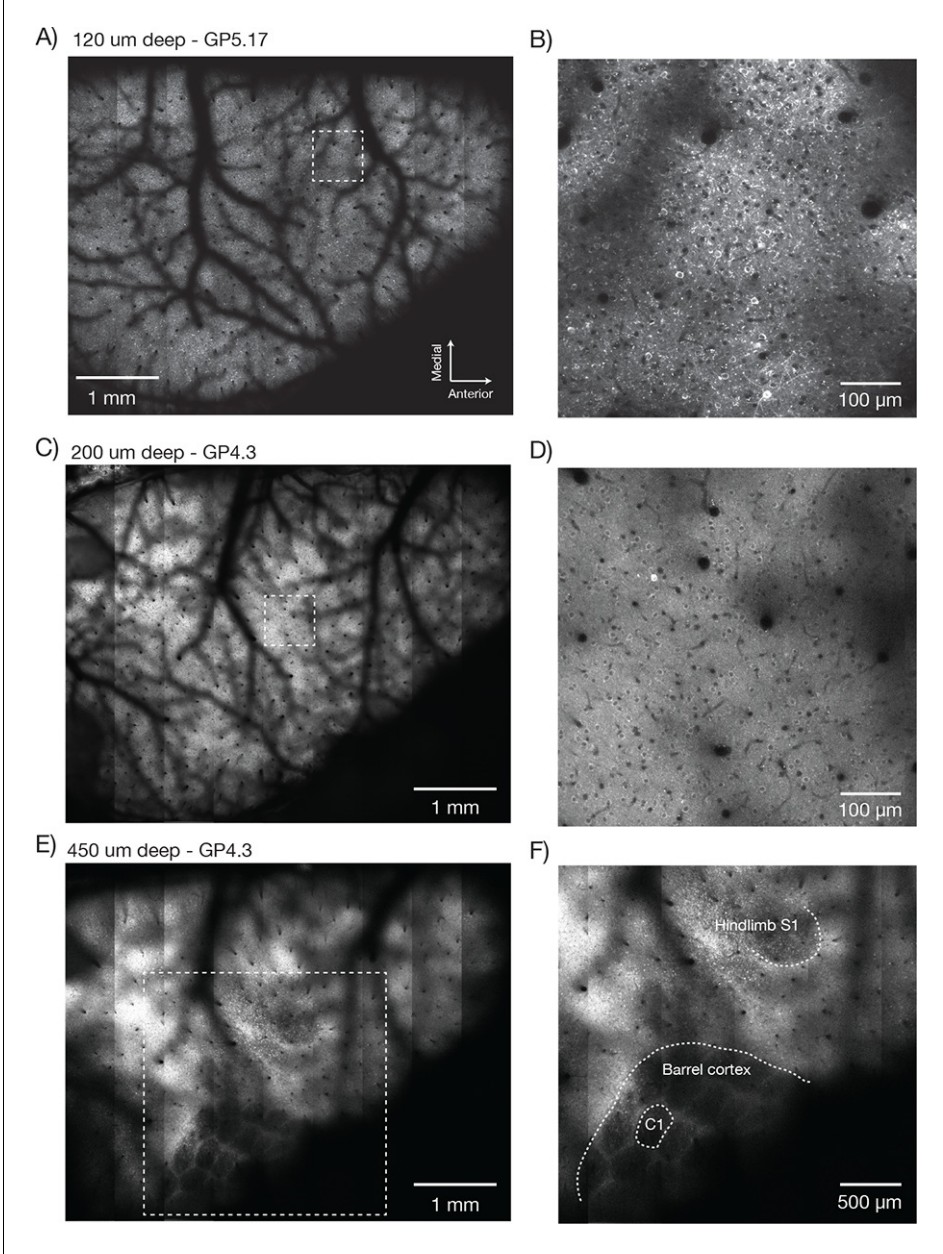

**Figure 7.** Imaging fluorescent proteins in anesthetized transgenic mice. (**A**) Low magnification image (one section, 120 μm below the dura). The mouse expressed GCaMP6f under the thy-1 promoter (GP 5.17 line). Scanning parameters as in *Figure 1Di*. Field curvature correction was enabled. Average of 30 scans; power 120 mW. (**B**) Higher magnification image (dashed box in **A**). Scanning parameters as in *Figure 1Dii*. (**C**) Low magnification image (one section, 200 μm below the dura). The mouse expressed GCaMP6s under the thy-1 promoter (GP 4.3 line). Scanning parameters as in Fig. 1Di. Field curvature correction was enabled. Average of 30 scans; power 120 mW. Data corresponds to *Video 1*. (**D**) Higher magnification image (dashed box in **C**). Single neurons with nuclear excluded GCaMP are clearly visible. (**E**) Low magnification image at a deeper focal plane (one section, 450 μm below the dura). Same mouse and scanning parameters as **C**); power 175 mW. (**F**) Higher magnification image (dashed box in **E**) showing major somatosensory areas in layer 4. These areas appear as regions with reduced fluorescence since GCaMP6s expression is minimal in layer 4 stellate cells.

The following figure supplements are available for figure 7:

**Figure supplement 1.** Overlay of a typical field of view with the reference atlas.

*Figure 7 continued on next page*

*Figure 7 continued*

**Figure supplement 2.** Counting nuclei.
**Figure supplement 3.** Dendritic images.

Deeper imaging, approximately 450 µm below the surface of the brain, revealed the outlines of the layer 4 barrels of the whisker representation area of the somatosensory cortex, and the lower limb and upper limb somatosensory areas (*Figure 7E,F*, *Figure 7—figure supplement 1A*; GP4.3). Layer 4 has decreased fluorescence because stellate cells do not express fluorescent proteins in these mice (*Dana et al., 2014*).

We next imaged GFP-labeled nuclei of excitatory neurons in emx1-Cre x lsl h2b-GFP mice (*Madisen, 2015*). We imaged the full FOV in 2 µm z-steps (up to 600 µm deep) and 1.2 x 1.3 µm$^2$ pixels (*Figure 7—figure supplement 1A*). We automatically detected nuclei in three dimensions using a peak finding algorithm (peak_local_max, scikit-image) (*Figure 7—figure supplement 1B*). We detected a total of 207,359 nuclei in the volume (*Figure 7—figure supplement 1C*). This is a lower bound on the number of neurons accessible for functional imaging in the 2p-RAM.

In mice expressing YFP in a sparse subset of pyramidal neurons (YFP-H) (*Feng et al., 2000*) we were able to resolve individual spines (*Figure 7—figure supplement 2*).

We next imaged neural activity in behaving mice running on a treadmill in tactile virtual reality (*Sofroniew et al., 2014*, *Sofroniew et al., 2015*). We imaged a 4.4 mm x 4.2 mm FOV at 1.9 Hz in layer 2/3 (*Figure 8—figure supplement 1A,B*, *Video 2*; GP4.3 mice). A simple correlation analysis revealed over 3000 active neurons (*Figure 8—figure supplement 1C,D*).

The 2p-RAM can link activity at the level of brain regions and single neurons. For example, low magnification movies showed activity across multiple brain regions including somatosensory, parietal, and motor areas (*Figure 8A*; GP5.17 mice). We correlated the average activity from an area (region 1) in somatosensory cortex with activity in the rest of brain (*Vanni and Murphy, 2014*) (*Figure 8B*). We then imaged region 1 and three additional, widely separated regions (each 600x600 µm$^2$) at higher speed (9.6 Hz) (*Figure 8C*, *Video 3*). Individual neurons showed fluorescence transients corresponding to trains of action potentials (*Chen et al., 2013*). We manually segmented individual neurons in each of the regions and computed the change in fluorescence (ΔF/F) over time (*Figure 8E*). We measured average activity in region 1 by computing the maximum of the ΔF/F traces across all neurons in that region at each time point. We then correlated this trace against all the ΔF/F traces of individual neurons (*Figure 8D,F*). Some neurons in the different areas showed high correlation with this average

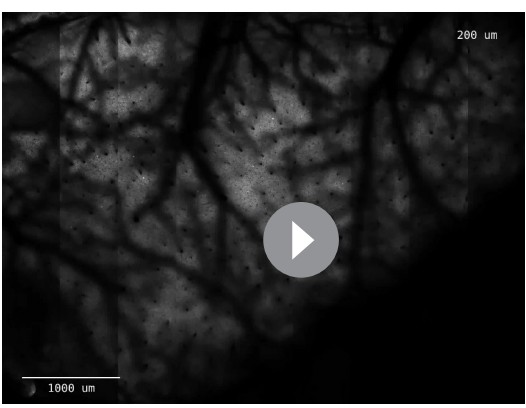

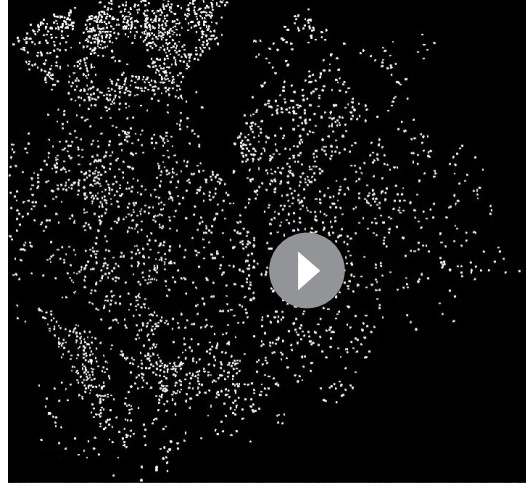

**Video 1.** Multi-resolution information in a 2p-RAM image stack. Same data as in *Figure 7C*.

**Video 2.** Functional imaging of a large field of view. Same data as in *Figure 8—figure supplement 1*. Movie data was smoothed and downsampled by a factor of two in time.

trace, whereas others showed uncorrelated activity (*Okun et al., 2015*). In this manner, the 2p-RAM can be used to relate activity across nearly the entire dorsal cortex with the activity of individual neurons in specific areas during behavior.

## Discussion

We designed a 2-photon random access mesoscope (2p-RAM) that allows rapid microscopic investigation of neural tissue over large imaging volumes. The 2p-RAM has a numerical aperture of 0.6 and an imaging volume corresponding to a ∅5x1 mm cylinder. The imaging volume is approximately 100-fold larger compared to other microscopes with comparable resolution.

The 2p-RAM is optimized for high-resolution in vivo imaging. Lateral scanning is performed using a resonant scanner in series with a xy galvo scanner. Rapid scanning is achieved by moving the fast resonant scan over the FOV using the galvo mirrors. Even faster and more flexible scanning could potentially be achieved using acousto-optic deflectors (*Duemani Reddy et al., 2008*) instead of resonant scanners.

Rapid axial scanning is critical for mesoscale functional imaging in the intact brain. This is because neurons of interest will generally be at different imaging depths in different brain regions. It is also often of interest to interrogate the same brain region at different imaging depths. In the 2p-RAM, axial scanning is based on aberration-free focusing using a remote mirror (*Botcherby et al., 2012*).

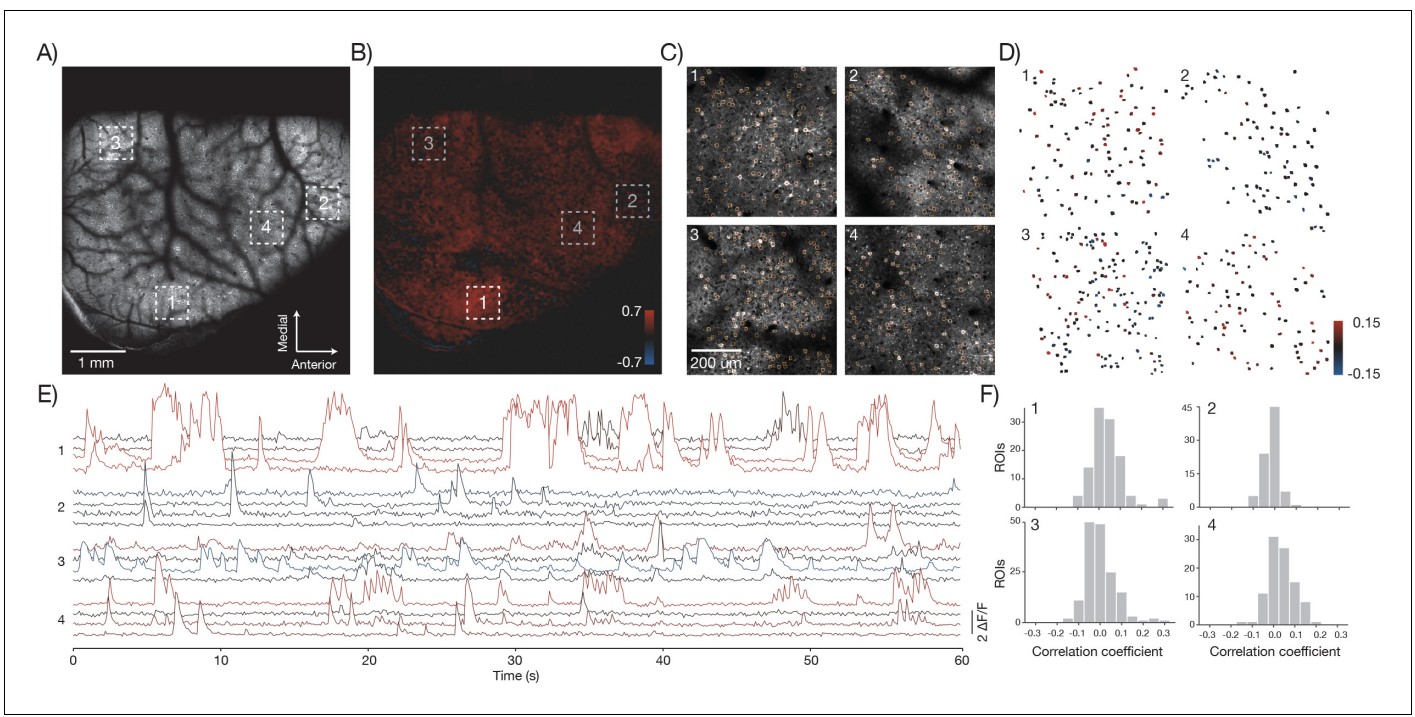

**Figure 8.** In vivo imaging of neural activity. (**A**) Low magnification image from a mouse expressing GCaMP6f under the thy-1 promoter (GP 5.17 line) (parameters described in Figure 1Div; 120 mW average laser power). Sampling rate, 4.3 Hz. (**B**) Cross-correlation map of the mean activity in region 1 with the activity in all pixels after smoothing with a 5x5 pixel boxcar filter. A highly correlated spot of activity is visible around region 1. (**C**) Four fields of view (corresponding to boxes in **A**) that were acquired at high resolution and frame rate (scan parameters describe in Figure 1Dv; power, 120 mW; sampling rate, 9.6 Hz). Regions of interest (orange) were manually drawn around individual neurons. (**D**) Each neuron from the four fields of view colored according to its correlation value with the average activity in region 1. The average activity in region 1 was computed by taking the maximum values across all the Δ *F* / *F* traces in that region. (**E**) Δ *F* / *F* traces for 16 neurons extracted from the four separate regions (four per region) colored according to their correlation coefficient shown in **D**. Sampling rate, 9.6 Hz. Data corresponds to *Video 3*. (**F**) Histograms of the correlation coefficients for the neurons in **D** in the four separate regions.

The following figure supplement is available for figure 8:

**Figure supplement 1.** Neural activity in a large field of view.

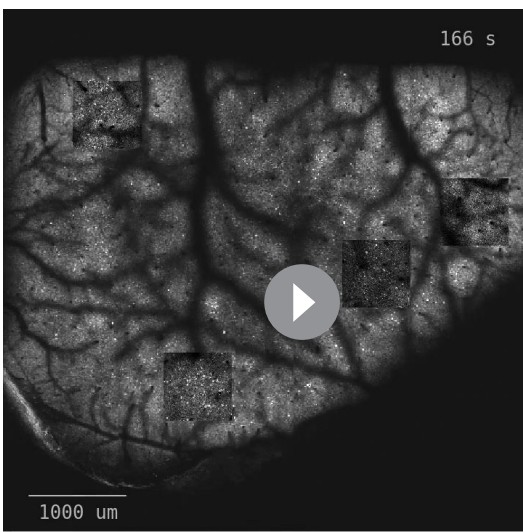

166 s

1000 um

**Video 3.** Low resolution imaging at low speed, followed by multi-area imaging at higher speeds. Same data as in *Figure 8A,C*. Low resolution movie data smoothed and downsampled by a factor of three in time. High resolution movie data median filtered in space with filter size two, and smoothed and down sampled by a factor of five in time.

The mirror is moved in a plane nearly conjugate to the specimen plane to move the focus. Aberrations are corrected by passing the beam through a RF objective that produces complimentary aberrations to the imaging objective. In published optical arrangements, the beam is scanned laterally first, and is then sent into the RF system (*Botcherby et al., 2012*). This design puts constraints on the RF objective. In our system the beam is passed through the RF system before lateral scanning. This configuration allowed the etendue, and thus size, complexity, and cost, of the RF objective to be much lower than that of the imaging objective. We also optimized the coatings and the element count of the RF objective to minimize power losses that are normally high in remote focusing systems. The optics subsequent to the RF objective could be made such that the remote focusing is invariant to lateral scanning, and thus diffraction-limited focusing can be achieved over a large volume with this arrangement. Alternative axial scanning techniques, such as rapidly moving the objective (*Göbel et al., 2007*), are impractical due to the large size of the 2p-RAM objective. Electrotunable lenses introduce spherical and chromatic aberrations (*Grewe et al., 2011*).

The 2p-RAM has relatively large field curvature (*Figure 6*). Rapid remote focusing can help to compensate for field curvature in real time during scanning. Online correction of field curvature can only be achieved for the movements produced by the slower galvo scanners. The movements of the resonant scanner are too fast to be fully compensated, because RF systems implemented with either galvos (*Botcherby et al., 2012*), voice coils (*Figure 6—figure supplement 1*), or piezos (unpublished observations) cannot match the resonant scanning bandwidth. Fast resonant scan lines are by necessity tilted, which is more pronounced close to the edge of the FOV. For many in vivo applications the residual field curvature is expected to be only a small inconvenience, because neurons of interest in different brain areas are typically in different focal planes.

The serial scan system allows for both low-magnification imaging across a large area and high-resolution imaging across smaller, but widely separated, areas. Low-magnification imaging can be combined with online analysis to identify brain areas with particular activity patterns of interest. Once these areas are found, high-resolution imaging can interrogate populations of individual neurons within those areas. The low-magnification image can be used to optimally place these smaller imaging regions around the vasculature.

We performed GDD compensation to maximize peak power delivered for a given average power. Larger peak power then can be used to maintain the same signal to noise ratio during faster scanning with smaller pixel dwell times. Optimal placement of imaging regions and faster scanning combined with smaller jump times between regions will allow for simultaneous tracking of tens of thousands neurons across the cortex at 10 Hz.

Microscopes with yet larger fields of view are possible. However, given fixed NA, larger fields of view correspond to larger objectives. Larger PMTs will be required to efficiently collect the fluorescence transmitted by these giant objectives.

There is considerable interest in more natural, unconstrained, and therefore more variable animal behaviors. Relating the underlying neural population activity to behavior will require single-trial analysis, which is limited by the number of neurons that can be recorded in multiple brain regions (*Churchland et al., 2007*). Mesoscale recording systems, such as 2p-RAM, allow recordings from thousands of neurons across multiple brain areas. These large numbers are critical to achieve single-trial decoding of neural activity patterns during behavior.

## Materials and methods

All surgical and experimental procedures were in accordance with protocols approved by the HHMI Janelia Research Campus Institutional Animal Care and Use Committee (IACUC 14–115). All mice were over 8 weeks old at the time of surgery. Two mice were females (both GP4.3; JAX 024275), and two were males (GP5.17; JAX 025393 and emx1-cre x lsl h2b-GFP; JAX 005628; JAX 023139, from lab of Josh Huang). Surgeries and anesthetized imaging were performed under isoflurane anesthesia (1–2%), while mice were maintained at 37°C with a heat blanket (*Huber et al., 2012*). Mice were given at least five days to recover from surgery before awake imaging experiments. Analysis code was written in Python and used Jupyter notebooks. Code, data, and CAD models are available at https://github.com/sofroniewn/2pRAM-paper.

## Acknowledgements

We thank Courtney Davis and Simon Peron for help with cranial window surgeries; David Tracy and Julie Bentley for help with optical design; Jon Arnold for mechanical engineering support; Jay Kumler, Dan Sykora, and Matt Falanga (Jenoptik) for help with optical design and testing; Jeff Brooker and Shane Patton (Thorlabs) for custom mechanical engineering; Aaron Kerlin and Vasily Goncharov for help with the remote focus system; Jeremy Freeman for help with analysis; Kaspar Podgorski, Arseny Finkelstein, Na Ji, Simon Peron, and Aaron Kerlin for comments on the manuscript.

## Additional information

### Competing interests

KS: Reviewing editor, *eLife*; Author on patent applications filed on behalf of Janelia Research Campus, HHMI. The 2p-RAM technology has been licensed commercially by Thorlabs Inc, and can be licensed separately for non-profit purposes. NJS, DF: Author on patent applications filed on behalf of Janelia Research Campus, HHMI. The 2p-RAM technology has been licensed commercially by Thorlabs Inc, and can be licensed separately for non-profit purposes. The other author declares that no competing interests exist.

### Funding

| Funder | Author |
|---|---|
| Howard Hughes Medical Institute | Karel Svoboda |

The funders had no role in study design, data collection and interpretation, or the decision to submit the work for publication.

### Author contributions

NJS, KS, Conception and design, Experiments, Analysis and interpretation of data, Drafting the article; DF, Conception and design, Optical and mechanical engineering, Experiments, Analysis and interpretation of data, Revising the article; JK, Conception and design, Software development, Revising the article, Acquisition of data, Analysis and interpretation of data

### Author ORCIDs

Daniel Flickinger, http://orcid.org/0000-0002-8361-3122
Karel Svoboda, http://orcid.org/0000-0002-6670-7362

### Ethics

Animal experimentation: All procedures were in accordance with protocols approved by the Janelia Research Campus Institutional Animal Care and Use Committee. IACUC 14-115.

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
