## [Decision Letter]

Thank you for submitting your article "A large field-of-view two-photon microscope with subcellular resolution for in vivo imaging" for consideration by *eLife*. Your article has been favorably evaluated by David Van Essen as the Senior editor and two reviewers, one of whom, Fred Rieke, is a member of our Board of Reviewing Editors.

The reviewers have discussed the reviews with one another and the Reviewing Editor has drafted this decision to help you prepare a revised submission.

This is a timely manuscript describing the design of a wide-field two photon microscope, capable of maintaining sub-micron lateral resolution over large fields of view (that would, e.g., cover multiple brain regions). The reviewers both felt the paper stood to make a substantial contribution by making the design of such microscopes accessible to other laboratories. Some key aspects of the microscope design are not discussed in sufficient detail to realize this goal, however. Furthermore, the writing is too technical in many places and could be made more accessible. See specific comments from the reviewers for details.

Reviewer #1:

This paper describes the design of a two-photon microscope that can image in a large field-of-view while maintaining sub cellular resolution. This is of obvious interest for many applications. Further, the focus on details of the microscope design and the particular choices made should be very helpful for others attempting to build similar microscopes. I am not a microscopy expert, so will restrict my comments to issues about presentation.

Organization: Figure 8 is very nice – that is really when you demonstrate the full power of the approach. It is worth considering moving that figure up and starting with it. You could use it to set up the design constraints. Starting with that figure could get many readers on the hook and interested in the more technical parts of the paper.

Reviewer #2:

General Assessment: Sofroniew et al. present a technical instrumentation manuscript describing a novel two-photon microscope with the following capabilities: 0.66 micron lateral resolution, 4.09 micron axial resolution, 5 mm x1 mm imaging volume, fast three dimensional scanning. Such capabilities will open up a new generation of in vivo imaging experiments in rodents by allowing for studies of cortical brain activity (with subcellular resolution) over a large portion of the cortex. The manuscript is timely and the microscope design should prove useful for many exciting studies examining the relationship between activity patterns in microcircuits simultaneously in spatially separate brain regions during behavior. As the authors point out, their new microscope is capable of recording from a ~100 fold larger imaging volume compared to other two-photon microscopes with similar resolution.

Overall the manuscript is direct and to the point; however, the writing seems rushed (there are numerous typos) and numerous details are missing. The following points should be addressed before publication:

1) One of the major goals of an instrumentation/technique manuscript, such as this one, should be to provide sufficient detail for the readers to reproduce the instrument/technique, but the authors have not provided sufficient detail for this to occur. For example, in the "Optical Design" section, the authors state that they will provide a detailed description of the optical designs elsewhere. This is confusing. Is there a more appropriate venue to publish such details other than the instrumentation paper presenting the microscope? Ideally, every element should be detailed (for a good example of appropriate level of detail, see Stirman and Smith 2014, 2015 bioRxiv), and at the very least, the designs for the custom optical components should be provided here. The authors should also comment on whether it is practical for others in the community to make such an instrument, or whether the cost is prohibitive.

2) In vivo resolution: The manuscript never explicitly states that resolution will be degraded, possibly severely, in vivo (for example, at the end of the first paragraph of the subsection “Resolution” it could be interpreted by some readers as a claim that the bead measurements will translate directly to in vivo conditions); while this degradation of PSF in vivo is obvious to experts in the field, it may not be obvious to many readers. Furthermore, it is not clear from the in vivo images how well signals from adjacent neurons (or neurons above/below each other) can be separated. For example, nuclear exclusion is not very clear in the figure images (contrary to what is stated in the second paragraph of the subsection “in vivo imaging”). Ideally, the authors would provide measurements of PSFs in vivo, provide better images (increased zoom) to see GCaMP nuclear exclusion, state explicitly that their PSF measurements are ex vivo and that in vivo PSFs will be larger, and preferably demonstrate signal separation between adjacent neurons.

3) Introduction, sixth paragraph: The authors’ argument that serial scanning of a single focal spot is superior to scanning multiple focal spots (multiplexing) seems to rely of the idea that the 200 mW power limit represents the total integrated power applied to the whole brain, but if multiple spots are separated by large (millimeter) distances, this power limit may not hold (as the authors’ suggest in the Discussion, fifth paragraph). If this power limit does not hold, does the authors’ serial vs. multiplexed scanning argument hold?

---

## [Author Response]

*This is a timely manuscript describing the design of a wide-field two photon microscope, capable of maintaining sub-micron lateral resolution over large fields of view (that would, e.g., cover multiple brain regions). The reviewers both felt the paper stood to make a substantial contribution by making the design of such microscopes accessible to other laboratories. Some key aspects of the microscope design are not discussed in sufficient detail to realize this goal, however. Furthermore, the writing is too technical in many places and could be made more accessible. See specific comments from the reviewers for details.*

*Reviewer #1:*

*This paper describes the design of a two-photon microscope that can image in a large field-of-view while maintaining sub cellular resolution. This is of obvious interest for many applications. Further, the focus on details of the microscope design and the particular choices made should be very helpful for others attempting to build similar microscopes. I am not a microscopy expert, so will restrict my comments to issues about presentation.*

Organization: Figure 8 is very nice – that is really when you demonstrate the full power of the approach. It is worth considering moving that figure up and starting with it. You could use it to set up the design constraints. Starting with that figure could get many readers on the hook and interested in the more technical parts of the paper.

We carefully considered your suggestion. However, in the end we feel that the logic flows better if we present the in vitrocalibration data first (i.e. the hard numbers on performance). The aficionados will also consider this ‘hard’ data more informative and important.

*Reviewer #2:*

*Overall the manuscript is direct and to the point; however, the writing seems rushed (there are numerous typos) and numerous details are missing. The following points should be addressed before publication:*

1) One of the major goals of an instrumentation/technique manuscript, such as this one, should be to provide sufficient detail for the readers to reproduce the instrument/technique, but the authors have not provided sufficient detail for this to occur. For example, in the "Optical Design" section, the authors state that they will provide a detailed description of the optical designs elsewhere. This is confusing. Is there a more appropriate venue to publish such details other than the instrumentation paper presenting the microscope? Ideally, every element should be detailed (for a good example of appropriate level of detail, see Stirman and Smith 2014, 2015 bioRxiv), and at the very least, the designs for the custom optical components should be provided here. The authors should also comment on whether it is practical for others in the community to make such an instrument, or whether the cost is prohibitive.

We have provided the part numbers of the custom optical elements. These parts can be ordered from their manufacturer (Jenoptik). We now also provide a supplementary figure containing the critical distances between the elements (Figure 2—figure supplement 1) and a link to the full 3D CAD model (URL). With this information it is possible to reproduce the instrument. Stirman et al do not give part numbers for their custom lenses, nor do they give sufficient information to exactly recreate them.

The cost of the custom optical elements is approximately $ 60,000. We are committed to make all details available, including machine shop ready files, bills of materials. This requires a significant effort in documentation. We will have this done by the end of the summer.

Note that we are currently not providing a description of the design principles and procedures that underlies the custom optics (the aforementioned Jenoptik parts). Instead we are preparing a paper on lens design, to be submitted to an optics journal. This is what we mean by ‘will provide a detailed description of the optical designs elsewhere’. Microscopists routinely describe new modes of microscopy with optical elements that are essentially black boxes (e.g. objectives and scan lenses). In most cases details of lens designs are simply not available from the manufacturers. Furthermore, we feel that the community interested in details of lens design overlaps little with the readership of *eLife*.

The future technical manuscript, focused on the optical design, will discuss the trade-offs made during the design process. It will contain details on the performance merit functions, the choice of glasses, and the tolerancing and testing procedures used to produce a workable microscope.

2) In vivo resolution: The manuscript never explicitly states that resolution will be degraded, possibly severely, in vivo (for example, at the end of the first paragraph of the subsection “Resolution” it could be interpreted by some readers as a claim that the bead measurements will translate directly to in vivo conditions); while this degradation of PSF in vivo is obvious to experts in the field, it may not be obvious to many readers. Furthermore, it is not clear from the in vivo images how well signals from adjacent neurons (or neurons above/below each other) can be separated. For example, nuclear exclusion is not very clear in the figure images (contrary to what is stated in the second paragraph of the subsection “in vivo imaging”). Ideally, the authors would provide measurements of PSFs in vivo, provide better images (increased zoom) to see GCaMP nuclear exclusion, state explicitly that their PSF measurements are ex vivo and that in vivo PSFs will be larger, and preferably demonstrate signal separation between adjacent neurons.

We have added language to clarify that the bead measurements were made in vitro. We now provide better images and supplemental movies, showing that nuclear exclusion is clearly visible across the field of view.

*3) Introduction, sixth paragraph: The authors’ argument that serial scanning of a single focal spot is superior to scanning multiple focal spots (multiplexing) seems to rely of the idea that the 200mW power limit represents the total integrated power applied to the whole brain, but if multiple spots are separated by large (millimeter) distances, this power limit may not hold (as the authors’ suggest in the Discussion, fifth paragraph). If this power limit does not hold, does the authors’ serial vs. multiplexed scanning argument hold?*

To avoid confusion we have removed the suggestion from the fifth paragraph of the Discussion. We estimate we could only achieve a modest 20% increase in total laser power because heat is dissipated over a large part of the brain. This is well below the 100% increase in power required for a two beam multiplexing approach to achieve comparable equivalent SNR.